# Improved Deep Q-Network for User-Side Battery Energy Storage Charging and Discharging Strategy in Industrial Parks

**DOI:** 10.3390/e23101311

**Published:** 2021-10-06

**Authors:** Shuai Chen, Chengpeng Jiang, Jinglin Li, Jinwei Xiang, Wendong Xiao

**Affiliations:** 1School of Automation and Electrical Engineering, University of Science and Technology Beijing, Beijing 100083, China; B20150284@xs.ustb.edu.cn (S.C.); B20160276@xs.ustb.edu.cn (C.J.); B20190280@xs.ustb.edu.cn (J.L.); G20209453@xs.ustb.edu.cn (J.X.); 2Key Laboratory of Knowledge Automation for Industrial Processes of Ministry of Education, School of Automation and Electrical Engineering, University of Science and Technology Beijing, Beijing 100083, China

**Keywords:** industrial parks, battery energy storage, deep Q-network, charging and discharging strategies

## Abstract

Battery energy storage technology is an important part of the industrial parks to ensure the stable power supply, and its rough charging and discharging mode is difficult to meet the application requirements of energy saving, emission reduction, cost reduction, and efficiency increase. As a classic method of deep reinforcement learning, the deep Q-network is widely used to solve the problem of user-side battery energy storage charging and discharging. In some scenarios, its performance has reached the level of human expert. However, the updating of storage priority in experience memory often lags behind updating of Q-network parameters. In response to the need for lean management of battery charging and discharging, this paper proposes an improved deep Q-network to update the priority of sequence samples and the training performance of deep neural network, which reduces the cost of charging and discharging action and energy consumption in the park. The proposed method considers factors such as real-time electricity price, battery status, and time. The energy consumption state, charging and discharging behavior, reward function, and neural network structure are designed to meet the flexible scheduling of charging and discharging strategies, and can finally realize the optimization of battery energy storage benefits. The proposed method can solve the problem of priority update lag, and improve the utilization efficiency and learning performance of the experience pool samples. The paper selects electricity price data from the United States and some regions of China for simulation experiments. Experimental results show that compared with the traditional algorithm, the proposed approach can achieve better performance in both electricity price systems, thereby greatly reducing the cost of battery energy storage and providing a stronger guarantee for the safe and stable operation of battery energy storage systems in industrial parks.

## 1. Introduction

With the integration of large-scale renewable energy equipment in a new power system, distributed wind power and photovoltaic power generation are affected by environmental factors, and the power generation output is random and volatile. The ready-to-use characteristics of traditional power systems are difficult to adapt to the development requirements of new power systems. Meanwhile, with the access of a large number of distributed new energy sources on the power user side, the accuracy of power load forecasting has also dropped significantly. Faced with the increasing peak-valley difference of power load, the short-term peak load demand results in the peak-shaving problems, such as high cost and low utilization rate. As a means of peak shaving and valley filling, the energy storage system has the characteristics of fast charging and discharging response speed. The reasonable configuration of energy storage can effectively alleviate the problem of voltage overruns and fluctuations caused by large-scale new energy grid connection [1,2,3].

Industrial parks have high electricity costs, rapid peak load growth, and strong demand for electricity savings. Therefore, energy storage-based peak shaving and valley filling, and peak-valley arbitrage are used to charge the grid at peak-valley price differences or during flat periods. Discharging in the peak period of electricity price, earning the electricity price difference, and obtaining the income of charging and discharging can significantly reduce the electricity purchase cost of enterprises. In addition, photovoltaic modules are laid on the roofs of all enterprises in the industrial park, and together with the energy storage system, a light storage microgrid system is constructed. That is, the photovoltaic system is used for power generation during the day and the energy storage system is used for discharging at night. Self-sufficiency can effectively reduce the operating costs of enterprises and respond to the call for energy conservation and emission reduction advocated by the state [4]. Therefore, in order to improve the consumption of new energy and save electricity costs in industrial parks, how to improve the performance of energy storage systems has become an important research direction.

With the rapid development of battery energy storage technology, multiple modes, such as centralized energy storage power stations and distributed battery energy storage, have emerged one after another. User-side battery energy storage refers to an electrochemical energy storage system that realizes the storage, conversion, and release of electric energy on the user side. The user-side battery energy storage system in the industrial park can achieve peak-shaving and valley-filling, and demand-side management of the internal load of the park can reduce the electricity cost of users in the park and promote the consumption of renewable energy in the park. Compared with residential load and commercial load, the peak-to-valley distribution of industrial load is more random, making peak shaving analysis more difficult [5,6,7]. Different from the unitary time-of-use electricity price, industrial users generally implement a diversified electricity price mechanism, and participating in peak shaving will bring multiple benefits [8]. The industrial user-side battery energy storage system can perform planned charging and discharging according to the difference in electricity prices at different times of the day, especially in areas with large peak-to-valley price differences in Jiangsu, Beijing, Shanghai, Tianjin, and Zhejiang, which has good application value. At the same time, the user-side battery energy storage system of the industrial park can respond to grid dispatch and assist the park to participate in the grid’s peak-shaving auxiliary service, which is the future development trend of the integrated energy system.

User-side battery energy storage mainly has problems, such as large initial investment, low return, and long investment recovery period. Many factors, such as its profit space, operation mode, and operation strategy, have been widely concerned. Compared with lead-acid batteries, lithium batteries have higher power density, capacity, and charging and discharging life, and do not require special cooling requirements. With the continuous improvement of the cost performance of lithium batteries, lithium batteries have been widely used as a backup power source, and have functions such as peak shaving and valley filling and power demand response to reduce power costs and increase revenue. The planning and operation of battery energy storage systems under peak shaving constraints was studied [9,10,11]. The modeling and optimal scheduling of demand response was introduced [12,13]. However, the above research do not consider the joint optimization of the battery energy storage system and the demand side response. A dispatch model with the goal of minimizing the difference between the load curve and the daily average load was established, and achieved the reduction of energy consumption and load peaks [14]. The above research is mainly aimed at the planning and economic evaluation of the off-battery energy storage system connected to the distribution network or microgrid.

Taking into account the power load characteristics of the industrial park, the city’s power supply price and the capacity of energy storage batteries, the study of adaptive charging and discharging technology to achieve peak-shaving and valley-filling of power supply has become the core problem of solving the park’s energy consumption optimization. With the development of battery energy storage technology, various battery charging and discharging management approaches have emerged one after another. According to different load scenarios, a rule-based online energy storage battery management approach is designed to deal with power emergencies [15]. A rule-based local controller to reduce computational complexity and meet the needs of rapid response was proposed [16]. An approach based on the Laypunov method to optimize the charging and discharging of energy storage batteries has been proposed to reduce electricity costs. In [17], a Laypunov-based optimization method was introduced to balance load and battery scheduling to minimize energy consumption costs. Literature [18] introduces the concept of pollution index, and uses the method of secondary planning to solve the energy consumption problem, which reduces the cost of energy consumption and improves the utilization rate of clean energy. A joint control method of wind power generation and energy storage based on model predictive control is proposed, which can well match the changes of power load and power supply [19].

Deep reinforcement learning (DRL) combines deep learning perception capabilities with reinforcement learning (RL) decision-making mechanisms. It is an intelligent method that is closer to human thinking [20]. Because RL has better adaptive capabilities, battery energy storage management technology based on RL has been widely studied. Some battery charging and discharging strategies based on RL have been proposed to optimize the charging and discharging strategies of energy storage batteries [21,22,23]. A state estimation algorithm for lithium-ion batteries based on RL is proposed, which achieves accurate estimation of the state of lithium batteries. However, this method is difficult for parameter modification [24].

With the rise in deep learning technology, DRL, which combines the advantages of both reinforcement learning and deep learning, has achieved good results in the field of artificial intelligence, and has improved the level of intelligence in areas such as autonomous driving, intelligent medical care, and industrial control. Deep Q-network (DQN) is a representative DRL algorithm based on value function [25,26]. A double DQN (DDQN) algorithm combined with double Q learning is proposed [27]. This algorithm uses an online Q-network estimation strategy, uses the target Q-network to estimate the *Q* value, and solves the problem of overestimation of the *Q* value. A priority experience replay DQN (PER-DQN) algorithm based on DDQN is introduced [28]. The algorithm considers the importance of different data, and often plays back important data. Wang et al. proposed the dueling DQN (DU-DQN) algorithm, which uses two independent estimators to represent the state value function and the state-related action function to improve the stability of the optimization process [29]. A dynamic evolutionary model based on the first kind Volterra integral equation is used to improve the performance of energy storage with renewable and diesel generation [30]. Combining the actual situation of lake Baikal region, a zero-emission hybrid AC/DC optimization model is introduced [31]. This framework is effective for grid community management and has high potential for CO_2_ reduction. Through the previous research and analysis, various battery energy storage charging and discharging strategies were proposed. However, these methods have some shortcomings in the sample priority update. Meanwhile, most of the existing methods have not studied the optimization of charging and discharging strategies for actual industrial electricity prices and energy storage battery operation. A novel cost-effective demand side management and peak power shaving based on vanadium redox flow battery are both demonstrated [32]. This energy management scheme is a scalable model for large-scale renewable energy integrated power systems. An optimized cooling system is proposed for kW scale Li-ion battery stack [33]. The battery discharging based on the cooling system has achieved good performance.

This paper mainly studies the charging and discharging strategy optimization technology of battery energy storage systems in industrial parks based on DRL. Based on the advantages of deep learning neural networks and reinforcement learning mechanisms, the optimal charging and discharging actions are selected, thereby reducing the electricity cost. Meanwhile, considering the temporal correlation of short-term power load, the temporal difference error (TD-error) is used to update the priorities of sequence samples in the experience pool. It facilitates the agent to obtain the preorder correlation samples with higher matching degree corresponding to charging/discharging conditions. Then, the modified rectified linear unit (M-ReLU) function is used to improve the performance of DQN.

The paper is arranged as follows. Section 2 gives a brief introduction to the battery energy storage system of the industrial park. Section 3 describes the definition of DQN and related technologies. In Section 4, the battery energy storage charging and discharging model and the optimization strategy based on DQN are constructed. Section 5 reports the experimental results. Finally, conclusions are given in Section 6.

## 2. Power Storage Management in Industrial Parks

The industrial park is a relatively independent power distribution infrastructure and an important part of the distribution network. It has become the trend of urban planning and development due to its highly intensive and highly shared advantages. China’s “carbon peak and carbon neutral” goal put forward higher requirements for the previous balance of production capacity and energy consumption in industrial parks. In order to ensure the normal operation of industrial parks, energy storage systems of a certain scale are usually configured as peak shaving power supplies and backup power supplies. The power supply system of the industrial park is mainly composed of the urban power supply network, diesel generators, and energy storage batteries. Among them, the industrial park energy management system is used for park power supply and energy storage battery charging and discharging management. Figure 1 shows a schematic diagram of the power supply system in the industrial park.

The urban power supply network provides electricity and electricity price information for the industrial park. Energy storage batteries are used for power storage to replace uninterruptible power supplies (UPS) as backup power sources, and for load regulation during peak power consumption. The industrial park energy management system controls the charging and discharging actions of energy storage batteries and the start and stop of diesel generators based on the information such as grid electricity prices, energy storage battery power, and office equipment workload, so as to reduce the energy consumption and electricity costs. The goal of energy storage battery charging and discharging strategy optimization is to maximize the benefits of charging and discharging, that is, to maximize the difference between the discharging revenue and the charging cost, and to maximize the savings in electricity costs.

The battery energy storage in the industrial park has two functions. One involves discharging during peak electricity consumption and charging when the price is low for saving electricity costs, and the other involves providing backup power when there is no power supply. The discharging power of the battery is affected by the discharging rate. Therefore, the optimal charging and discharging actions of energy storage batteries are directly affected by the battery capacity, charging and discharging currents, and the electricity prices.

Rate capacity effect and recovery effect are two important indicators that affect battery performance. The empirical formula for lead-acid batteries is the Peukert equation, and the battery capacity changes with the discharging current [34]. The larger the discharging current, the smaller the battery capacity. The Peukert formula satisfy
(1)$$I^{n}t=K,$$
where *I* is the discharging current and *n* is the Peukert constant (1.15 ≤ *n* ≤ 1.42). *K* is a constant related to the amount of the active material in the battery, and represents the theoretical capacity of the battery.

Taking into account the different chemical characteristics of lithium batteries and lead-acid batteries, the researchers improved the above formula. During the charging and discharging process, we need to reasonably choose the charging and discharging actions according to the remaining battery power. The estimation methods of battery remaining capacity mainly include an open-circuit voltage method, a coulomb counting method, an artificial neural network method, and a Kalman filter method [35]. This paper chooses coulomb counting method to estimate the remaining capacity of the battery. At the moment *t*, the calculation method of the remaining battery capacity can be expressed as
(2)$$E_{st}\left(t\right)=E_{st}\left(0\right)-\sum_{t^{\prime }=1}^{t-1}P_{st}\left(t^{\prime }\right)\times D,$$
where *E_st_*(0) is the battery power at the initial moment. *P_st_*(*t′*) is the charging and discharging power at time *t′*, and *D* is the duration.

## 3. Basis of Deep Q-Network

RL is a machine learning algorithm developed based on the theories of animal learning, parameter disturbance adaptive control, etc. DRL combines the perception ability of deep learning with the decision-making ability of reinforcement learning, which continuously interacts with the environment in a trial and error manner, and obtains the optimal strategy by maximizing accumulated rewards [36,37,38].

### 3.1. Reinforcement Learning

RL is an extensively used technique to train autonomous agents through experimentation. First, an action that affects the environment is chosen, then the agent observes how much that action collaborated to the task completion through a reward function. It is about taking suitable action to maximize reward in a particular situation. It is employed by various software and machines to find the best possible behavior or path it should take in a specific situation.

RL has been widely used in the fields of automatic control, engineering modeling, and model optimization. Its core idea is to achieve the maximum return or achieve specific goals through learning strategies during the interaction between the agent and the environment, so that the agent has the ability to make optimal decisions. Considering that the transformation process of reinforcement learning between environments is very complicated, in order to simplify the reinforcement learning modeling problem, the Markov decision process (MDP) is proposed to describe and model the reinforcement learning process [39]. The principle of reinforcement learning is shown in Figure 2. The specific description is as follows

Agent: It is an assumed entity which performs actions in an environment to gain some reward.

Environment: A scenario that an agent has to face.

Reward: An immediate return given to an agent when he or she performs specific action or task.

State: State refers to the current situation returned by the environment.

The goal of reinforcement learning is to maximize the cumulative reward, and the future trend of rewards needs to be considered when calculating rewards. Cumulative reward is defined as the weighted sum of rewards from time *t* to the end of the learning process, and its mathematical expression is as follows
(3)$$R_{t}=\sum_{t^{\prime }=t}^{T}\gamma^{t^{\prime }-t_{r_{t}}},$$
where *γ* ∈ [0, 1] is a constant, called the discount coefficient, which is used to evaluate the impact of future rewards on cumulative rewards.

The state action function *Q^π^*(*s*, *a*) represents the execution of action *a* in the current state *s*, and loops to the end of learning according to the strategy *π*. The cumulative return of the agent can be expressed as
*Q^π^*(*s*, *a*) = *E*[*R_t_*|*s_t_* = *s*, *a_t_* = *a*, *π*].(4)

For all state action sets, if the expected return of a strategy *π*^*^ is greater than or equal to the expected return of other strategies, then the strategy is the optimal strategy. In fact, it is possible that multiple optimal strategies share the same state action function
*Q^*^*(*s*, *a*) = max*_π_ E*[*R_t_*|*s_t_* = *s*, *a_t_* = *a*, *π*].(5)
This action function is called the optimal state action function and follows the Bellman optimal equation, which can be expressed as
*Q^*^*(*s*, *a*) = *E*_s′~s_[*r* + *γ*max_a′_*Q*(*s*′, *a*′)|*s*, *a*].(6)

Theoretically, the *Q* value function can be solved by iterative Bellman equation. In practical applications, methods such as neural networks and linear functions are used to approximate the state action value function, which realizes the integration of deep learning and reinforcement learning, and further promotes the development of DRL.

### 3.2. Deep Q-Network

DRL combines the perception capabilities of deep neural networks and the adaptive mechanism of reinforcement learning to form end-to-end perception and control. In general, DRL mainly involves two methods based on value function and strategy gradient. Deep neural network is used to approximate the value function and strategy gradient, and achieves good results in games, autonomous driving, and robot control [40,41].

DRL based on value function is usually used in low-dimensional and discrete action spaces. In the model of this paper, the agent selects battery charging and discharging actions every time period. The battery-related state dimension is low. The number of discrete charging/discharging actions is small, and the DRL method based on the value function can realize the optimal action selection.

DQN is a kind of DRL, which is a combination of deep learning and Q learning. DQN uses a deep neural network to estimate the action value (*Q*) function. The DQN model is composed of a current value network, a target value network, an error function, a playback memory unit, and other parts. In order to solve the problem of instability or even non-convergence of the neural network approaching action value function, DQN uses an empirical playback mechanism and target network to solve these problems. Figure 3 shows the network structure of DQN.

The DQN algorithm defines the loss function as the variance between the target value and the predicted value. The formula is
(7)$$L=E\left[{\left(r+\gamma \underset{a^{\prime }}{\max}Q\left(s^{\prime },a^{\prime },\theta^{-}\right)-Q\left(s,a,\theta \right)\right)}^{2}\right].$$

First, the target network will be frozen after several steps of training. Then, every *N* steps copy the current value network parameters to the target value network parameters, thereby stabilizing the training process and making the model easier to converge. Considering that the traditional DQN algorithm has the shortcomings of over-estimation of *Q* value, weak directivity, and poor stability, some improved DQN methods have been proposed [42,43,44].

## 4. Charging and Discharging Strategy Optimization of Energy Storage Battery

### 4.1. System Modeling

The charging and discharging control of the energy storage battery can be regarded as the selection of different discrete current values. Considering that the charging and discharging actions of the energy storage battery are taken every 15 min, we propose to use DQN and its improved method to manage the charging and discharging decision of the energy storage battery. The energy storage battery system provides backup power when the power grid is out of power, and provides power supply services during peak load periods, thereby reducing power consumption in the park and promoting energy conservation and emission reduction. According to the park load, battery capacity, and the backup power support time when the power grid loses power, the battery capacity used to provide backup power and the battery capacity used to reduce peak and valley loads are jointly determined. The mathematical relationship is as follows
(8)$$E_{st}=E_{total}-P_{ec}\times T,$$
where *E_st_* is the battery capacity used for peak shaving and valley filling, *E_total_* is the current capacity of the energy storage battery, *P_ec_* is the power of the park’s electrical equipment, and *T* is the support time of the backup power supply when the grid is out of power. The size of *T* is determined according to the actual needs of the operations of the digital park, with the general value 10~30 min. The agent makes optimal charging and discharging decisions based on the available capacity *E_st_*, the charging and discharging current *I_a_*, and the electricity price *V_pr_*. Next, we define the charging and discharging optimization problem of the energy storage system.

The state space is *S* = {*s* = [*E_st_*, *E_total_*, *P_ec_*, *V_pr_*]^T^| *E_st_* ∈ ***E_st_***, *E_total_* ∈ ***E_total_***, *P_ec_* ∈ ***P_ec_***, *V_pr_* ∈ ***V_pr_***}, where *E_st_*, *E_total_*, *P_ec_*, and *V_pr_* represent the set of available energy, total energy, power of electrical equipment, and electricity price in a limited state, respectively.

The action space is *A* = {*a* = *P_st_*| *P_st_* ∈ ***P_st_***}. ***P_st_*** represents the power group that the battery charging and discharging under different charging and discharging currents in a finite state, and the value of *P_st_* is a finite value within the range of [−*P_st_^max^*, *P_st_^max^*]. *P_st_* = 0 indicates that the energy storage battery system is not charged or discharged, *P_st_* > 0 indicates that the energy storage battery system is discharged, and *P_st_* < 0 indicates that the energy storage battery system is charged.

The reward function is the reward that the system obtains by choosing action *a* in state *s*. This paper defines the ideal return in the *t*-th time period as a reward function. The discharging reward function and the charging reward function are expressed as *R* = (*V_pr_*(*t*) − *V_pr_^min^*) × *P_st_* × *T* and *R* = (*V_pr_^max^* − *V_pr_*(*t*)) × *P_st_* × *T*, respectively. Among them, *T* is the duration, *V_pr_*(*t*) is the electricity price at time *t*, *P_st_* is the charging and discharging power of the energy storage system. *V_pr_^max^* and *V_pr_^min^* are the highest electricity price and the lowest electricity price, respectively.

### 4.2. Primary Parameters of DQN

The optimized state space of the energy storage battery system in the industrial park is a one-dimensional vector with a small dimension. The Q-network and the target network in this paper use multilayer neural networks. The number of neurons in the input layer is the dimension of the state space data vector, the number of neurons in the output layer is the number of actions in the action space, and the number of neurons in the hidden layer is greater than the number of neurons in the input and output layers. The optimal number of neurons and layers are selected according the actual experiment, and choose the ReLU as the action function [45].

### 4.3. Charging and Discharging Strategy Optimization Based on Improved DQN

In this section, we propose a new charging and discharging strategy to improve the efficiency of battery energy storage. This strategy optimizes the model of DQN sample priority update and the ReLU function.

#### 4.3.1. Charging and Discharging Optimization Algorithm

The industrial park energy storage battery system takes into account the functions of energy storage and UPS. The UPS battery is in fully charge state for a long time continuously, with less charge/discharge times. However, energy storage batteries need to be charged and discharged frequently. In order to make full use of the battery life cycle, the paper divides the batteries into two groups, which are alternately used for charging and discharging and UPS, thereby increasing the battery life.

In order to realize the optimal charging and discharging control of energy storage battery, this paper uses DQN and its improved algorithm to control the charging and discharging actions. The purpose is to ensure that the energy storage battery obtains good performance in a complex working environment, and to minimize the power consumption cost of the park. Algorithm 1 describes the battery charging and discharging management algorithm based on DQN.
**Algorithm 1** Battery charging and discharging optimization algorithm**Require:** electricity price *V_pr_***Ensure:** earned money *money*1: *money*, *step*, *changedStep* = 02: **repeat**3:  *r* = getRandom();4:  **if**
*r* < *ϵ*
**then**5:     selet *a_t_* randomly;6:  **else**7:     select *a_t_* = *argmax_a_Q*(*s_t_*, *a*|*θ*);8:  **end if**9:  *ϵ* = *ϵ* − △*ϵ*10:      execute charging/discharging action *a_t_*, and get reward *r_t_* and new state *s*_*t*+1_;11:      store (*s_t_*, *a_t_*, *V_pr_*, *s*_*t*+1_) in replay memory *D*;12:      sample random minibatch of transitions from *D*;13:      calculate accumulative reward by target Q-network with parameters *θ*^−^;14:      perform a gradient decent learning on Q-network with parameters *θ*;15:      **if**
*step*/*N* == 0 **then**16:      update target Q-network parameters with Q-network parameters;17:      **end if**18:      **if**
*changedStep* + + > *M*
**and** isFull(*E_st_*) **then**19:      switch to next battery group;20:      *changedStep* = 0;21:      **end if**22:      calculate earned *money*;23: **until** (step = = MaxStep)24: **return**
*money*;

In Algorithm 1, lines 2–9 are randomly selected actions or selected actions based on the maximum *Q* value. Lines 10–17 are DRL training. Lines 18–21 are battery pack switching, the purpose is to maintain the balance of battery life. The algorithm is based on the *ε*-greedy algorithm, which realizes the exploration and utilization of the optimal actions. In the exploration stage, the algorithm selects the action with the highest *Q* value with a probability of 1 – *ε*, and selects other actions with the probability of *ε*. In the initial stage of training, *ε* can be set to a larger value so that the algorithm can explore as many actions as possible. During the training process, the exploration rate *ε* gradually decreases as the number of iterations increases, and the algorithm preferentially selects the action with the largest *Q* value, which has good convergence. After passing the training phase, the algorithm directly selects the action with the highest *Q* value.

The above is the battery charging and discharging management algorithm structure based on DQN. Other improved DQN models only modify the Q-network structure, training method or memory recall mechanism, and the overall framework and battery switching mechanism remain unchanged.

#### 4.3.2. Improved DQN Based on TD-Error and Modified ReLU Function

In the power supply system, the peak load presents a certain law in continuous time, which belongs to a non-stationary random time series, and has an obvious power consumption trend and a time series correlation. In PER-DQN, priority is measured by the TD-error between the target *Q* value and the actual *Q* value. PER-DQN first uses the priority *p* of the sequence sample *i* to calculate the probability $P\left(i\right)=p_{i}^{\sigma }/\left(\sum_{k}p_{k}^{\sigma }\right)$. that is sampled in *k* samples, and then calculates its corresponding sampling weight $\omega_{i}=\left(\frac{1}{N}*\frac{1}{P\left(i\right)}\right)\mu$ to effectively compensate for the error. The exponents *σ* and *μ* represent the priority and the sampling weight, respectively (range ∈ [0, 1]). *N* is the capacity of the experience pool.

Through the analysis of the training process, when the priority of a set of sequence samples is very small, the priority of the samples before the decision point will not be adjusted basically, making it difficult to update the TD-error stored in the experience pool in time. The parameters of the Q-network are continuously updated as the number of iterations increase, while the experience pool only updates the TD-error of the training samples. The number of such samples only occupies a small part of the experience pool capacity, and the TD-error of most samples cannot change with the changes of the Q-network, resulting in a deviation between the priority of the sample in the experience pool and the actual priority. In actual situations, the priority of sequence samples may appear zero, which is likely to cause “priority collapse” and intelligent weight sampling. In order to solve the above problems, this paper uses the constant *c* and the parameter *η* to improve the performance of sample priority update. The *p_i_* = |TD-error| + *c* method is used to avoid the situation where the priority is zero. The updated priority *p_i_* is denoted as
*p_i_* = max(|TD-error| + *c*, *λ* × *p_i_*),(9)
where *c* is a small constant. *λ* ensures that priority is decreases slowly. In order to solve the problem of prioritized sequence experience replay, ζ is introduced to optimize the update strategy. *i* is the starting point, replaying the priority of sequence sample data within a window range *L*. The update rules are as follows
*p_i_*_−1_*=* max(*p_i_*_−1_ + *c*, *ζ* × *p_i_*),(10)
*p_i_*_−2_*=* max(*p_i_*_−2_ + *c*, *ζ* × *p_i_*),(11)
*p*_*i*−(*L*−1)_ = max(*p*_*i*−(*L*−1)_ + *c*, *ζ* × *p_i_*),(12)
where *ζ* is the attenuation coefficient, which indicates the degree of priority influence between adjacent decision points. According to experimental tests, *c*, *λ*, *ζ*, and *L* are set to 10^−4^, 0.73, 0.65, and 10, respectively.

For DRL algorithms, the real-time update of parameters is the basis for the agent to find the optimal timing scheme. The traditional ReLU function can effectively solve the problem of gradient disappearance, and significantly improve the performance of neural network. The ReLU function can be expressed by *f*(*x*) = manx(0, *x*). At *x* < 0, the calculation result does not converge. Therefore, the disadvantage of mean shift still exists. In order to solve this problem, we use the tanh function to optimize ReLU function. The modified function M-ReLU is realized by
(13)$$f\left(x\right)=\{\begin{array}{c} x,\;\;\;\;\;\;\;x\geq 0\\ \varphi \tanh\left(x\right),\;\;\;\;\;\;\;\;\mathrm{otherwise} \end{array},$$
where *φ* can be understood as the slope of curve. Compared with the ReLU function, the mean value of M-ReLU (*x* < 0) is close to 0, which avoids the appearance of mean shift. At the same time, the non-saturation of *f*(*x*) can alleviate the problem of neuronal death, and has better robustness to input changes.

## 5. Experiment and Analysis

### 5.1. Experimental Environment and Settings

We implement the DQN model using Python and TensorFlow. The hardware processor and memory are CPU: Inter(R) Core(TM) i7-6700 CPU @ 3.40 Ghz and 8G. The experiment is equipped with 100 lithium batteries, each with a capacity of 100 Ah. The batteries are divided into two groups; 50% of the battery capacity is used for UPS emergency power supply and the other 50% of the battery capacity is used for energy storage charging and discharging. Two groups of batteries take turns to charge and discharge energy storage to increase the battery life. Table 1 shows the detailed parameters of the lithium battery. This paper uses a DQN-based method to control the charging and discharging of energy storage batteries, and compares the performance with traditional algorithms DQN, DDQN, PER-DQN, and DU-DQN.

This experiment defines seven actions, including three charging actions, three discharging actions, and one non-charging/discharging action. Considering that the charging and discharging efficiency of lithium batteries is affected by the rate capacity effect and the recovery effect, the increase in the current will result in a decrease in the charging and discharging efficiency. In order to simplify the complexity of the model, the experiment sets several sets of typical empirical data on the current, voltage, and other parameters of the charging and discharging actions. Table 2 shows the charging and discharging actions and parameters set in this experiment.

This experiment uses the electricity price information *V_pr_BA_* of the Baltimore Company of the United States and the electricity price information *V_pr_BJ_* of Beijing as the experimental data. The electricity price of Baltimore Company belongs to the real-time electricity price, which has a strong correlation with the electricity supply and demand in each time period. Figure 4 shows the electricity price trend of *V_pr_BA_* on a certain day [46].

In fact, electricity prices fluctuate every day. This experiment adds −20~20% random fluctuations on the above data to form the training data of the DRL model. *V_pr_BJ_* belong to peak-to-valley prices. Each day is divided into peak time period, stable time period and low time period, and the price of each time period is fixed. Due to the tight power supply demand, the electricity prices of 11:00~13:00 and 16:00~17:00 during the summer from July to August belong to the higher peak electricity prices. In Figure 5, the non-summer and summer price curves of *V_pr_BJ_* are shown [47].

### 5.2. Performance Analysis of Traditional DQN Model

This experiment uses *V_pr_BA_* and *V_pr_BJ_* to train the DQN model, respectively. The experiment performs the selection of charging and discharging actions every 15 min, and there are a total of 96 selections of actions every day.

The performance of the DQN model is closely related to the structure and parameters of the neural network. The learning rate, reward discount rate, training data storage space, batch size, and step size of the DQN model are set to 0.0005, 0.95, 10,000, 32, and 400, respectively. The step size of 400 indicates that the target network parameters are updated every 400 steps. At the initial moment of training, the value of *ε*-greedy algorithm *ε* is set to 1, and a decreasing exploration is performed in steps of 0.0005 until the value of *ε* becomes 0. The first 500 steps of training are used for data collection. Reinforcement learning is not performed. DRL starts after 500 steps. In the scenario of battery charging and discharging management, the fully connected neural network is suitable for modeling parameters such as electric energy, power, current, and price. The experiment uses different neural network parameters (such as the number of layers, the number of neurons, etc.) to optimize the performance of the algorithm to find the optimal neural network parameters. The charging and discharging revenue of energy storage batteries is defined as the revenue from discharging minus the cost of charging. Figure 6 shows the benefits of the DQN model based on *V_pr_BA_* training and verification.

According to the experimental results shown in Figure 6, the algorithm converges after about 200 days of training, can choose the correct action, and can maintain the subsequent time gain at USD 1.9. Therefore, the DQN model can achieve stable performance, especially in the case of fluctuations in electricity prices, as it has strong robustness and adaptability.

Figure 7 shows the returns of DQN model based on *V_pr_BJ_*. The experimental results show that the model becomes stable after 200 days of training, and approximately CNY 82 of income per day can be obtained. In the summer months of July and August, Beijing implements peak electricity prices, and the DQN model can quickly capture the changes in electricity prices and make better action choices. This also reflects the advantages of the DRL algorithm in terms of adaptation.

The design of the neural network has a greater impact on the performance of the DNQ algorithm. The appropriate neural network structure and parameters can more accurately estimate the relationship between battery capacity, power, and electricity prices. Otherwise, over-fitting or under-fitting is prone to occur. We compare the 30-day average returns of different neural network layers and neurons. According to the results shown in Table 3, when the DQN model uses 3 layers of neural networks and 25 neurons, the revenue optimization of the three types of electricity price data can obtain better support. It can be seen from Table 3 that the neural network used in this experiment has good adaptive ability and can achieve good results.

In order to further analyze the specific conditions of the selected actions of the DQN model, we compare the fluctuations of the electricity price of a day and the corresponding actions in Figure 8. Experimental results show that the DQN-based model can select the correct charging action when the electricity price is low, and the correct discharging action when the electricity price is high. In order to obtain better income, the experimental system chose to charge twice a day, and choose non-charging/discharging during the time period when the electricity price is in the middle.

### 5.3. Improved DQN Model Performance Analysis

In order to analyze the performance of different algorithms, we compare the daily average returns of DQN, DDQN, DU-DQN, PER-DQN, with the proposed approach. Figure 9 depicts the returns of different models in *V_pr_BA_* data.

Through the comparison of experimental results, the performance of the four modified DQN algorithms is better than the traditional DQN algorithm. The performance of DDQN, DU-DQN, PER-DQN, and proposed method increased by 9.4%, 10.4%, 12.6%, and 13.3%, respectively. In order to analyze the performance of the DQN-based algorithm under *V_pr_BJ_* data, we did the same experiment.

It can be seen from Figure 10 that both the proposed method can achieve the best performance. In this paper, the same reinforcement learning parameters are used under different price systems. Figure 11 shows the average returns of the five DQN-based algorithms under the conditions of *V_pr_BA_*, *V_pr_BJ_*, and *V_pr_BJ_S_*. From the experiment, it can be seen that the proposed approach can achieve the best benefits under *V_pr_BA_* and *V_pr_BJ_S_*. The DDQN achieves better returns than PER-DQN based on *V_pr_BJ_*. In this paper, the same DRL parameters are used in different electricity price systems. Experiments show that the algorithm based on DQN has good adaptive ability, and the algorithm has good versatility and robustness under different electricity price systems.

### 5.4. Cost Analysis

According to the price of lithium batteries and the charging and discharging life, the cost of purchasing 100 lithium batteries requires CNY 100,000. According to the scheme proposed in this paper, the electricity bill can be saved by about CNY 98 per day, and the electricity bill can be saved by CNY 35,770 a year. The cost of lithium batteries can be recovered within 3 years. At this time, each battery has performed approximately 1200~1300 charging and discharging operations, which accounts for about 60% of the battery’s charging and discharging life. The solution of combining lithium battery energy storage and backup power is obviously better than the UPS solution that is only used for power backup in terms of economy. With the maturity of lithium battery production technology and the continuous decline of prices, this paper proposes a plan to recover the cost of lithium batteries faster, bring more energy storage benefits to industrial parks, and further reduce electricity costs. The analysis of the experimental results show that the technical solution of integrating energy storage and backup power has significant economic benefits, and is conducive to the access and utilization of renewable energy, such as wind and light.

## 6. Conclusions

This paper proposes an optimization algorithm for charging and discharging energy storage batteries based on DRL. The modified DQN model is used to control the charging and discharging of energy storage batteries, which achieves peak-shaving and valley-filling of electricity load in industrial parks and reduces electricity costs. DRL has the characteristics of simple parameter setting and strong compatibility, which can be applied to various electricity price systems. The simulation experiment shows that the proposed method can give a good charging and discharging strategy for the energy storage battery system. At the same time, the proposed method can greatly promote the cost recovery of energy storage batteries. With the integration of new energy sources, such as photovoltaics and wind power, future work can be carried out on optimal energy consumption management strategies based on DRL to improve energy efficiency in industrial parks, reduce electricity costs, and increase the consumption capacity of new energy.

## Figures and Tables

**Figure 1 entropy-23-01311-f001:**
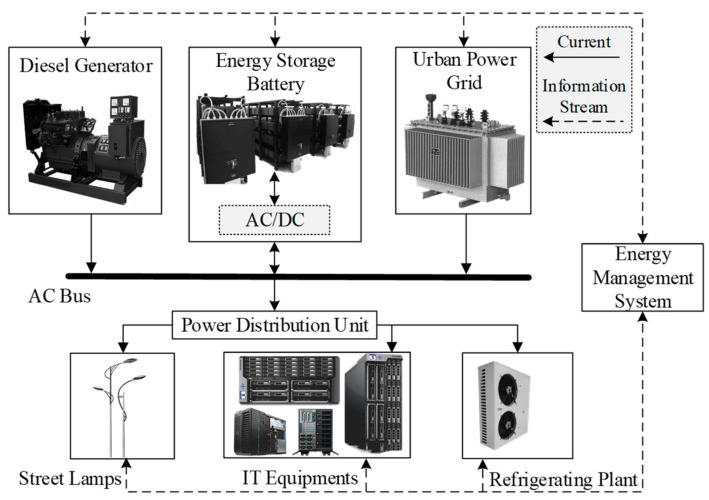
Power supply system of industrial parks.

**Figure 2 entropy-23-01311-f002:**
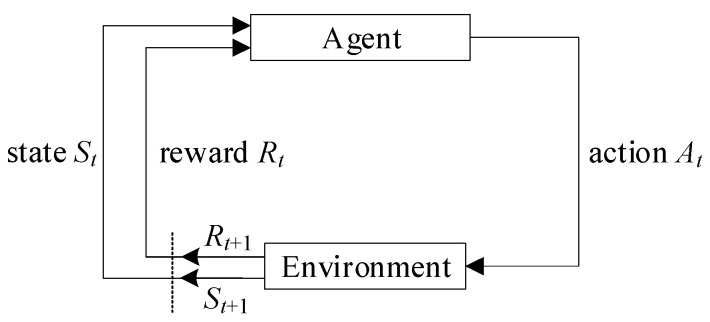
Schematic diagram of reinforcement learning principle.

**Figure 3 entropy-23-01311-f003:**
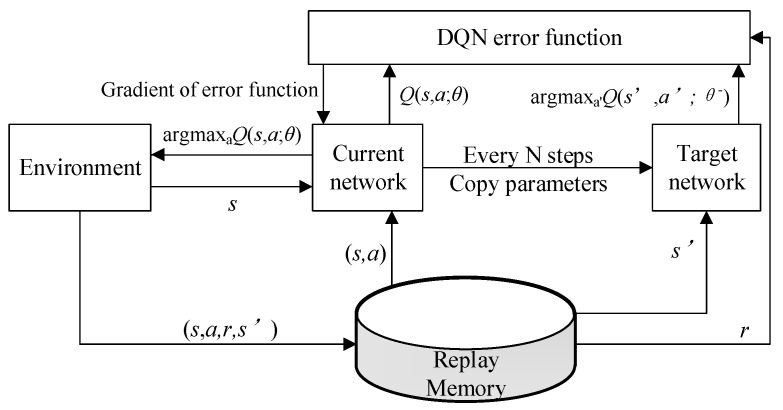
DQN network structure.

**Figure 4 entropy-23-01311-f004:**
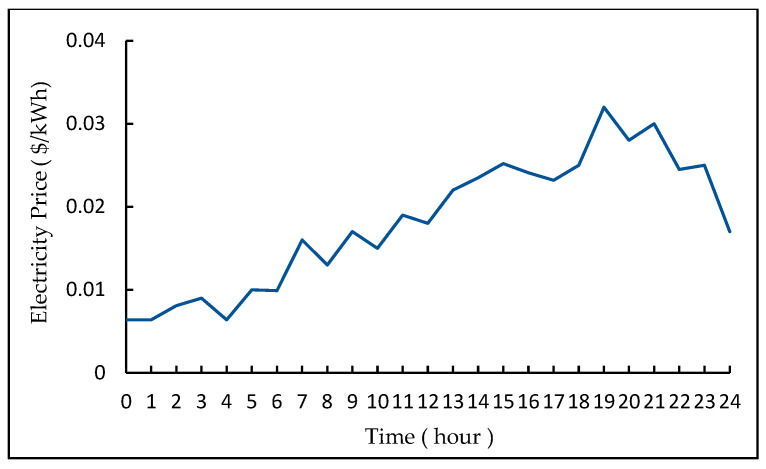
The schematic diagram of *V_pr_BA_*.

**Figure 5 entropy-23-01311-f005:**
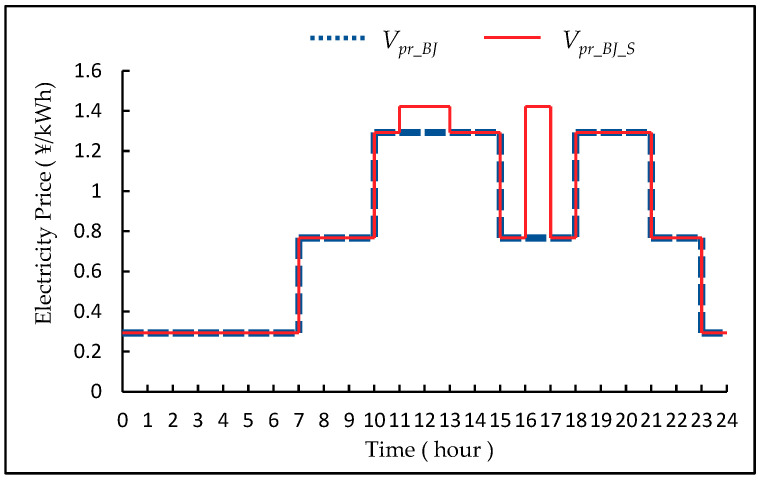
The schematic diagram of *V_pr_BJ_* and *V_pr_BJ_S_*.

**Figure 6 entropy-23-01311-f006:**
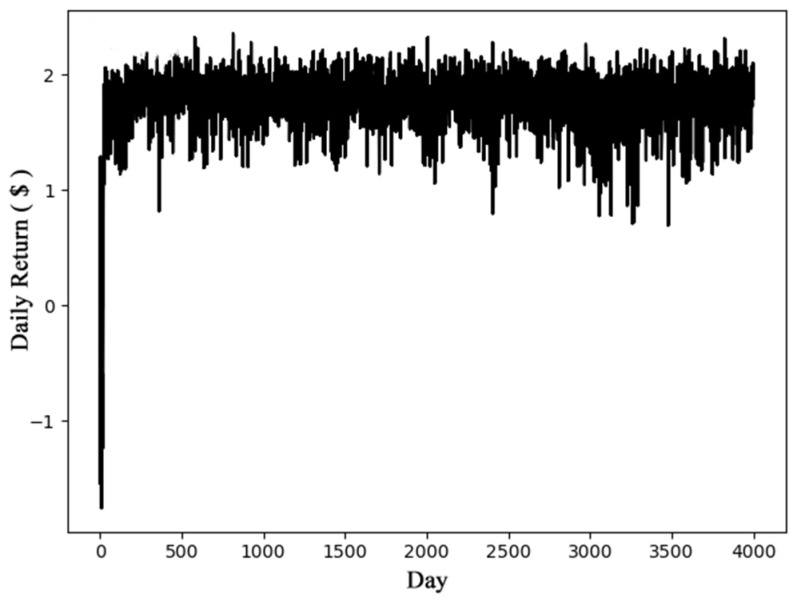
Daily return based on DQN and *V_pr_BA_*.

**Figure 7 entropy-23-01311-f007:**
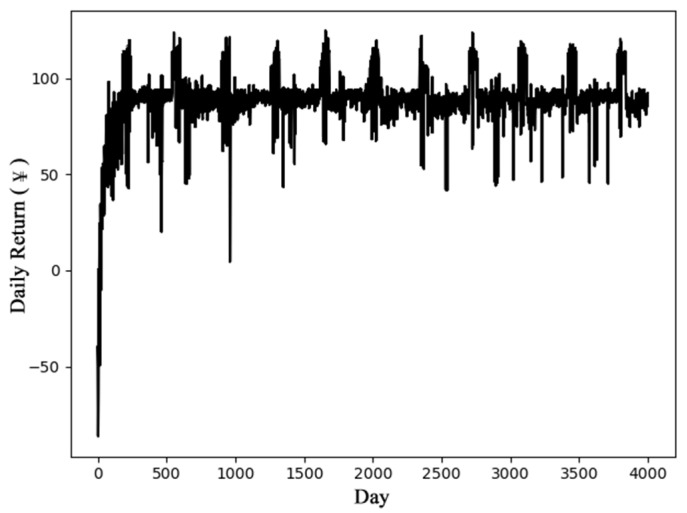
Daily return based on DQN and *V_pr_BJ_*.

**Figure 8 entropy-23-01311-f008:**
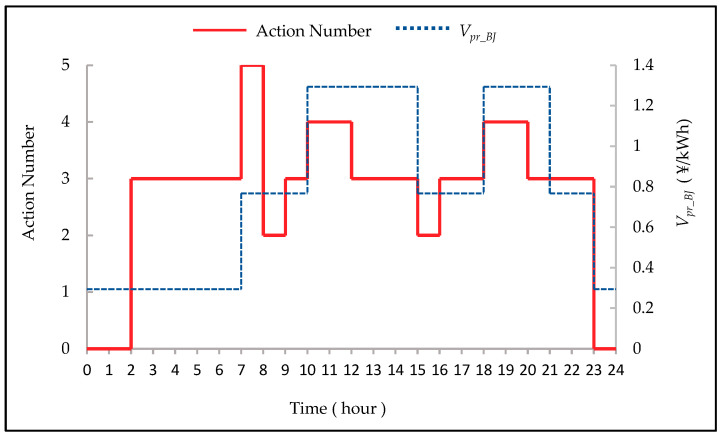
Comparison of charging/discharging actions and correspding *V_pr_BJ_*.

**Figure 9 entropy-23-01311-f009:**
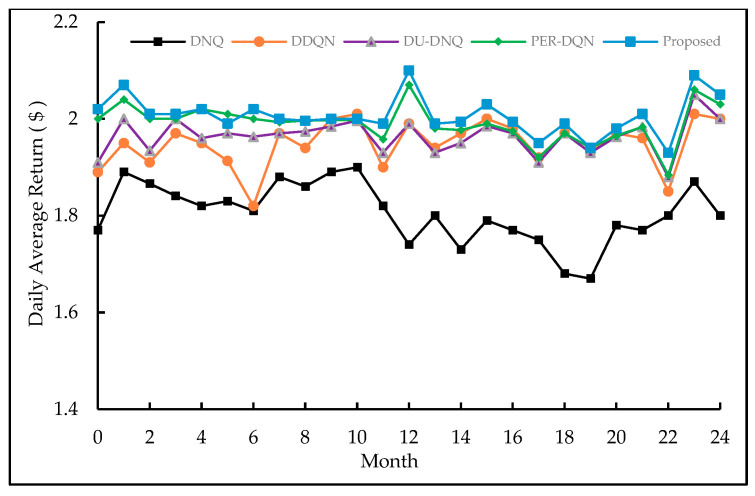
Performance comparison of four DNQ-based methods based on *V_pr_BA_*.

**Figure 10 entropy-23-01311-f010:**
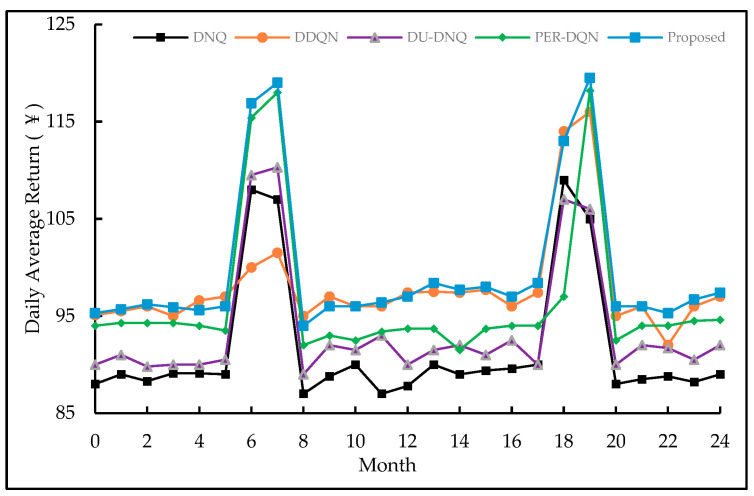
Performance comparison of four DQN-based methods based on *V_pr_BJ_*.

**Figure 11 entropy-23-01311-f011:**
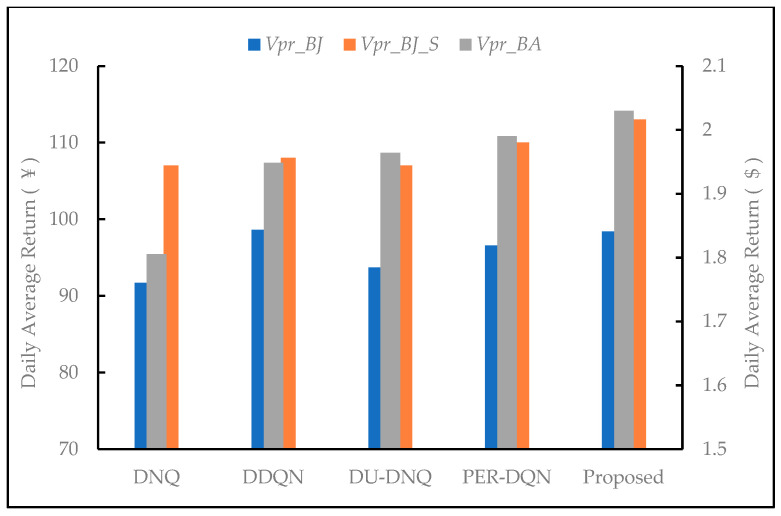
Daily average returns of DQN-based algorithms.

**Table 1 entropy-23-01311-t001:** The parameters of the lithium battery.

Category	Indicator	Detail
electrical specification	Nominal Voltage	12.8 V
	Rated Capacity	100 Ah
	Power	1280 Wh
	Cycle Life	>2000 cycles @1C 100%DOD
	Charging Efficiency	100% @0.5C
	Discharging Efficiency	96~99% @1C
Standard charging	Charging Voltage	14.6 ± 0.2 V
	Charging Type	CC/CV
	Charging Current	50 A
	Maximum Charging Current	100 A
	Charging Limited Voltage	15.6 V ± 0.2 V
Standard discharging	Continuous Current	100 A
	Maximum Pulse Current	120 A (<3 s)
	End of Discharging Voltage	8 V

**Table 2 entropy-23-01311-t002:** The parameters of charging and discharging actions.

No.	Action	Voltage (V)	Current (A)	Efficiency (%)
0	Charging	14.6	50	100%
1	Charging	14.6	80	95%
2	Charging	14.6	100	90%
3	Inaction	-	-	-
4	Discharging	12.8	50	100%
5	Discharging	12.8	100	96%
6	Discharging	12.8	110	90%

**Table 3 entropy-23-01311-t003:** Average return of different neural network models.

No.	Number of Neural Network Layers	Number of Neurons	Average Return		
			*V_pr_BA_* (USD)	*V_pr_BJ_S_* (JPY)	*V_pr_BJ_* (JPY)
1	2	25	1.85	78.94	106.69
2	2	35	1.77	93.28	112.97
3	2	45	1.84	93.74	105.32
4	3	25	1.95	94.55	114.22
5	3	35	1.94	88.29	107.73
6	3	45	1.93	90.46	109.98
7	4	25	1.87	91.39	114.28
8	4	35	1.96	84.87	107.10
9	4	45	1.85	93.06	111.25

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
