# Peer review of "Improved Deep Q-Network for User-Side Battery Energy Storage Charging and Discharging Strategy in Industrial Parks"

_entropy, 2021, doi:10.3390/e23101311_

Round 1
Reviewer 1 Report
Mathematical theory and models development for efficient battery energy storage systems control is essential field of contemporary industrial mathematics. Since recently due to the progress of HPC the approximation theory based on deep learning (including reinforcement learning) became very popular in different fields. The main contribution of this paper is employment and testing battery energy storage charging/discharging model and the optimization strategy based on DQN.
Suggestions:
- Please outline the main benefits of proposed approach comparing with other architectures proposed for storage systems control using DQN. The benefits must be clearly outlined in abstract, introduction and in conclusion
- Authors may wish to outline other dynamical models of storage systems, see [1,2].
- The readers will appreciate nomenclature.
[1] D. Sidorov, D. Panasetsky, N. Tomin, D. Karamov, A. Zhukov, I. Muftahov, A. Dreglea, F. Liu, Y. Li, Toward Zero-Emission Hybrid AC/DC Power Systems with Renewable Energy Sources and Storages: A Case Study from Lake Baikal Region”, Energies, 13:5 (2020), 1226.
[2] D. Sidorov et al., "A Dynamic Analysis of Energy Storage With Renewable and Diesel Generation Using Volterra Equations," in IEEE Transactions on Industrial Informatics, vol. 16, no. 5, pp. 3451-3459, May 2020, doi: 10.1109/TII.2019.2932453.
Reviewer 2 Report
The article proposed a new charging and discharging mode through an improved deep Q network for updating the priority of sequence samples and train a deep neural network. Good work in the field of battery management. Hope to see the implementation of this model in a real-time system.
Reviewer 3 Report
Author investigated “Improved Deep Q Network for User-Side Battery Energy Storage age Charging and Discharging Strategy in Industrial Parks”
This work is interesting however needs some modification.
- In this work new algorithm was developed for battery energy storage. Now the question how this model is applicable to every battery? Is it suitable for only certain types or universal? For Stationary battery now a days vanadium is most promising. Can this model satisfy for that? Also now EV application is more and storage is one of the criteria. Can this model work with that too?
- How this RL method was trained?
- Use equation no for all the mathematical formula. Line 220 what is r in trt? Seems the notation is not correctly written.
- Figure 4: A reference is required for the electricity price
- Yes electricity price is very much variable e.g. here UK electricity price is shown. This can be used a reference.
https://doi.org/10.1002/ente.202100199
https://doi.org/10.3390/en13215695
- Figure 6 and 7 the unit of price should be kept in single unit and keep universally accepted one. Dollar pound euro is ok
- Inclusion of the algorithm is good approach.
- What is the justification to include the details of CPU? Also how the batteries were divided for charging and storage? Don’t you think it seems same and storage is only possible while its charged through charging methods.
- Why electricity price is high in the night and during the peak day-mid day period.
- Figure 5 energy price seems logical.
- More justification is required for Figure 6. Currently it seems noise
Round 2
Reviewer 1 Report
Most of the suggestions have been considered.
In line 295 authors outlined "Considering that the traditional DQN algorithm has the shortcomings of over-estimation of Q value, weak directivity, and poor stability, some improved DQN methods have been selected". Pls add references here. Moreover, this must be clarified because loss function selection is of the principal steps. Why loss function was selected using (7)?
Author Response
Comments and Suggestions for Authors:
Most of the suggestions have been considered.
In line 295 authors outlined "Considering that the traditional DQN algorithm has the shortcomings of over-estimation of Q value, weak directivity, and poor stability, some improved DQN methods have been selected". Pls add references here. Moreover, this must be clarified because loss function selection is of the principal steps. Why loss function was selected using (7)?
Response: Thank you for your constructive and helpful suggestion. We added some references in line 297. This loss function is proposed in the traditional DNQ. This paper does not conduct research on the loss function, and the original loss function is directly adopted..
Reviewer 3 Report
All comments are addressed with sufficient answer.
Author Response
All comments are addressed with sufficient answer.
Response: Thank you very much for your approval of this article.